# Optimization Experiment of Production Processes Using a Dynamic Decision Support Method: A Solution to Complex Problems in Industrial Manufacturing for Small and Medium-Sized Enterprises

**DOI:** 10.3390/s23094498

**Published:** 2023-05-05

**Authors:** Simona Skėrė, Aušra Žvironienė, Kazimieras Juzėnas, Stasė Petraitienė

**Affiliations:** 1Faculty of Mechanical Engineering and Design, Kaunas University of Technology, 44249 Kaunas, Lithuania; 2Department of Applied Mathematics, Kaunas University of Technology, 44249 Kaunas, Lithuania

**Keywords:** smart production planning, production optimization, Industry 4.0, dynamic decision support method

## Abstract

In the industrial sector, production processes are continuously evolving, but issues and delays in production are still commonplace. Complex problems often require input from production managers or experts even though Industry 4.0 provides advanced technological solutions. Small and medium-sized enterprises (SMEs) normally rely more on expert opinion since they face difficulties implementing the newest and most advanced Industry 4.0 technologies. This reliance on human expertise can cause delays in the production processes, ultimately, impacting the efficiency and profitability of the enterprise. As SMEs are mostly niche markets and produce small batches, dynamics in production operations and the need for quick responses cannot be avoided. To address these issues, a decision support method for dynamic production planning (DSM DPP) was developed to optimize the production processes. This method involves the use of algorithms and programming in Matlab to create a decision support module that provides solutions to complex problems in real-time. The aim of this method is to combine not only technical but also human factors to efficiently optimize dynamic production planning. It is hardly noticeable in other methods the involvement of human factors such as skills of operations, speed of working, or salary size. The method itself is based on real-time data so examples of the required I 4.0 technologies for production sites are described in this article—Industrial Internet of Things, blockchains, sensors, etc. Each technology is presented with examples of usage and the requirement for it. Moreover, to confirm the effectiveness of this method, tests were made with real data that were acquired from a metal processing company in Lithuania. The method was tested with existing production orders, and found to be universal, making it adaptable to different production settings. This study presents a practical solution to complex problems in industrial settings and demonstrates the potential for DSM DPP to improve production processes while checking the latest data from production sites that are conducted through cloud systems, sensors, IoT, etc. The implementation of this method in SMEs could result in significant improvements in production efficiency, ultimately, leading to increased profitability.

## 1. Introduction

Small and medium-sized enterprises (SMEs) play a crucial role in the contemporary manufacturing industry. These businesses typically have fewer than 250 employees and can be subdivided into microenterprises (fewer than 10 employees), small enterprises (10 to 49 employees), and medium-sized enterprises (50 to 249 employees). Companies in such segments represent the majority of the existing business sites [1]. Despite their size, SMEs provide a diverse range of products, including niche products that may not be produced by larger companies; thus, they are often characterized by their flexibility and adaptability. However, having a wide range of products and an uncertain number of orders makes planning complicated. At the moment, a large number of decisions in SMEs are linked to expert opinions, yet since SMEs often struggle to attract and retain skilled workers owing to competition from larger companies, this origin of decisions needs to have advanced and computerized support. However, the nature of the SMEs’ production processes and their limited resources often mean that they cannot easily adopt the most advanced Industry 4.0 technologies, such as autonomous robots, automatic production lines, and advanced virtual, or augmented reality systems [2]. Connected to this problem, a method was created, and its capabilities are presented in this article. Our developed and proposed planning method would give the possibility for improvement in production planning without any magnificent investments. The core idea was to create such methods that would be easily adaptable, yet would cover the technical side of the company (such as available machinery, real data about in-stock materials, and their quantities) as well as human workforce problems—frequent changes of employees, lack of suitable experts, sick leaves, etc. High diversity in orders leads to another problem that employees only know specific operations and not everyone can be covering all tasks, thereby creating production stops. After instigating numerous research, it was confirmed that it is quite rare to find a method that would involve not only technical data but also the skills of employees and their absence or speed of working. For example, in [3] a smart production planning and control system was described but no attention was provided to the mentioned problem—human workforce. As well, Ref. [4] presented a decision support system for dynamic job-shop scheduling using real-time data with simulation but it only mentioned workstations with their operations but no employees who would perform the tasks in the workstations. Another study showed the results of the increased productivity of operators but did not mention how to select the employees, what their capabilities were to perform the specific tasks, or even how to solve situations when this employee was absent [5]. Based on these and other reviews, DSM DPP shows novelty in the field of production planning when human force is involved. One of the main components for this method to work is real-time data and there are many highly useful technologies in this field that do not require significant investments and can be easily adapted by SMEs. For example, the use of the Industrial Internet of Things (IIoT), sensors, blockchains, and data analytics can present the newest data, which is needed to optimize manufacturing processes and improve productivity based on the method. Similarly, cloud-based systems can store and analyze large amounts of data, enabling real-time decision-making without requiring significant investment in IT infrastructure. SMEs must adapt to the new Industry 4.0 technological advancements to stay competitive and effective in today’s fast-paced and ever-changing manufacturing industry. The adoption of Industry 4.0 technologies can help to optimize production processes, reduce costs, and improve the quality of products, ultimately, leading to increased competitiveness and profitability.

The article is presented in six sections. Section 1 is the Introduction, Section 2 covers the real-time data technologies that are adaptable for SMEs (IIoT, automatic quality control, and blockchains), Section 3 presents the required data for investigation and case study, Section 4 provides the main results, Section 5 offers a discussion on the existing limitations of this method, and Section 6 delivers the conclusions. Observations and data collection from the case study company were the key aspects of the whole research.

## 2. Real Time Data Technologies for SMEs

### 2.1. Industrial Internet of Things (IIoT)

Industry 4.0 brought IoT (Internet of Things) to everyday life and it refers to the network of physical devices, vehicles, and other objects that are embedded with sensors, software, and connectivity, which allows them to collect and exchange data with other devices and systems over the internet. IoT is typically used to refer to consumer applications, such as smart homes, wearable devices, and connected cars [6]. We cannot even imagine our lifestyle without this technology. The concept of IoT (Internet of Things) can be traced back to the 1990s when researchers at MIT began exploring the idea of “Things That Think” and “Everyware”. However, the term “Internet of Things” was first coined by Kevin Ashton in 1999, and the development of IoT technologies and applications really took off in the 2000s with the proliferation of wireless connectivity, cloud computing, and affordable sensors and devices [7].

After several years, this term was expanded and the concept of IIoT came to life—IIoT (Industrial Internet of Things) focuses specifically on the use of IoT technologies and concepts in industrial settings. IIoT involves the integration of sensors, data analytics, and connectivity into industrial machinery, equipment, and processes to enable real-time monitoring, predictive maintenance, and optimization of production processes. The term IIoT (Industrial Internet of Things) was first introduced by General Electric (GE) in 2012 to describe the use of IoT technologies in industrial settings [8]. Since then, IIoT has gained traction in various industries such as manufacturing, energy, transportation, and healthcare, as companies seek to leverage data and connectivity to improve efficiency, productivity, and safety. In summary, while IoT and IIoT share many of the same underlying technologies and concepts, they are differentiated by their specific application contexts. IoT is focused on consumer applications, while IIoT is focused on industrial applications.

Here are some examples of IIoT (Industrial Internet of Things) applications in the industrial field:Predictive maintenance: IIoT sensors can be used to monitor the health of industrial equipment and predict when maintenance is needed, reducing downtime, and increasing efficiency. For example, later in this research a method will be presented to solve production planning problems and one of the common issues is equipment stops, which could be prevented while getting up-to-date information.Quality control: IIoT sensors can be used to monitor and control the quality of industrial processes, ensuring that products meet specifications and reduce waste. Section 2.2 will describe the wider possible IIoT usage in quality checking.Supply chain management: IIoT technologies can be used to track and optimize the movement of goods and materials through the supply chain, reducing costs and improving efficiency. This could be used internally in the factory to know which material is currently in which production stage. The lack of materials is another open problem in production processes.Energy management: IIoT sensors can be used to monitor and optimize the use of energy in industrial facilities, reducing costs and improving sustainability. This is a highly focused topic in these energy crisis times. Saving time in production with created methods and involving advanced technologies ensures the best results can be reached.Worker safety: IIoT sensors can be used to monitor and improve worker safety in industrial settings, detecting and preventing accidents and injuries. During the pandemic, companies started using different prevention techniques, even for checking the health of employees—temperature sensors, face scanning to prevent mask usage, etc. [9].

All of these mentioned examples could be easily adapted, do not require huge investments, and would be a quick update for any sized SME. This shows the huge potential of this technology and the need for it.

### 2.2. Automatic Quality Control

Increasing in popularity are optical methods to observe displacements, velocities, and strains of surfaces. One of the biggest advantages of these methods is the ability to implement non-contact methods and receive information instantly [10]. This is the key factor for our method. Information could be transferred immediately; thus, the response would be much faster.

The laboratory of composite materials and adaptive structures presented optical devices to measure complex structures. They are an accurate, reliable, and easy-to-use device that examines the deformation behavior of complex three-dimensional surfaces and are made from innovative materials. Figure 1 shows the principle of the device—the light is split into two coherent beams. One is called “reference”, whereby optical fiber is wrapped around a piezo tube, which can be controlled to induce phase shifting. The second beam enters a switch box to modify the ESPI (electronic speckle pattern interferometry) configuration [11]. These two light fields interfere, and the resulting light field has random amplitude, phase, and intensity; thus, is a speckle pattern. If the object is deformed, the distance between the object and the image will change; hence, the phase of the image speckle pattern will change. Any inadequacy of such a chain will provide an automatic response for the whole process [12].

Many examples could be found of how optical checks are used in production. To perform high-quality control only visual human inspection is not enough since it relies on the experience of the inspector [13]. Furthermore, automation cannot be reached with such checks. Indeed, Ref. [14] presents how a novel visual sensing system for keyhole TIG is designed. The deep learning framework allowed the system to sort welding as good, incomplete, misaligned, undercut, or burnt through from just an image, taken with a camera, and the system could decide the quality of the welding. The received accuracy was not lower than 0.985.

Combining two main subsystems—image acquisition and image processing—an automatic visual inspection (AVI) system could be implemented. To implement such a system, some overall configuration must be followed. A lightning system generates light in a specific manner to illuminate the object and receive better-quality images. This illuminated scene is projected onto the sensor of a digital camera to create a digital image. This image is sent to a processing platform that controls all inspection processes. The processing platform analyzes and processes the acquired images to give the final inspection information. These are the basic descriptions of an AVI system, which also incorporates multiple software solutions to control different elements of the system (cameras and lighting system) [15]. Such automatic defect detections would lead to further automatization—automatic production order release for rejected quantities. Since the camera would not only be able to detect scratches, surface imperfections, and oiled surfaces but also categorize the part that has the inspection failure; thus, the system could automatically release production orders for the specific operation—surface cleaning or grinding. Further, having this data in the system allows the statistics of rejected parts to be collected and improvements to be made based on the information of which operation has failed the most. Here, neural networks would be used. To detect any defects, the system must be trained, and its sample flow chart is presented in Figure 2.

The process starts with the need to resize images to reduce the load of the system [16] since the general requirements of any AVI system are fast, cost-effective, and reliable systems [15]. Then, the system would do two checks—oil and scratch detections. If a part has any oil surplus on the surface, the part is rejected, and a new production order is released for a cleaning operation. If the part is clean but has surface defects or scratches, then, the production order is released for a grinding operation. This automatic quality inspection would provide data in real-time. As the created method checks the availability of the materials, this would provide an understanding of if there is enough raw material to make full orders in case some percentage is rejected.

### 2.3. Blockchains

Blockchains have emerged as a promising technology with the potential to revolutionize production processes by enhancing transparency, security, and efficiency [17,18,19]. In the context of production, blockchains can be used to track the movement of goods, monitor equipment performance, and manage supply chain relationships [20,21]. By leveraging the inherent characteristics of blockchains, such as immutability and transparency, manufacturers can ensure the accuracy and tamper-proofing of production data, thereby mitigating the risk of errors and fraudulent activities. Furthermore, the use of blockchain technology can facilitate seamless communication and collaboration between different stakeholders in the production process, such as suppliers, manufacturers, and distributors, thus, enhancing the overall efficiency and effectiveness of production processes. As such, blockchains hold significant potential to optimize production processes and improve the overall performance of manufacturing firms [22,23].

In the created method, a dynamic decision support module is proposed. It is enabled when no straightforward solution can be found for missing material, employee, or equipment failure. This module is based on data from other companies as subcontractors—their working hours, capacity, lead times, price, etc. This technology does not require high investments but would lead to better production performance.

## 3. Testing of DSM DPP at SME: Case Study

This section presents experimental research with the created method. The data were taken from a metal processing company, which is in the SME section. The company produces furniture components from metal and does not have many integrated I 4.0 technologies. Most work is conducted in relation to the opinions of the experts. The company has two shifts, which work 5 days a week. The term, employee-centered company could be used to describe this company—no automatic lines, robots, or conveyors are used in it. For this research, information was conducted on previously produced orders, and then, checked how optimization would affect them in the timeline. In total, 29 production orders were uploaded within the program.

Generally, one working day or shift could be presented, as in Figure 3. This sequence of steps follows for nearly every manufacturing company—a production order is released, then, the check of materials, equipment, and employees is made to begin the final step and initiate the production task.

This section will be subdivided into subsections based on Figure 3. Each part of the production will be described based on the created method before the results are presented in Section 4.

### 3.1. Production Order

For the created method, to be initiated, each production order must have specific initial information. As shown in Table 1, different information is required but it is basic information that does not require specific knowledge. This research is created with data from 16 production orders, with Table 1 presenting 6 of them. The created algorithm plans the production based on the importance of each different parameter.

Operations data and the time spent on each operation are shown in Table 2. Each product might have a unique sequence of operations and a different time of operation.

Columns for both tables are described in Table 3. Even though this is only a small amount of data and the whole information is very important, the number one task is to eliminate unnecessary and negligible values. It is checked which parameters have the most influence on order sequencing and if it is noticed that some have no influence, they are eliminated. As an example, subcontracting is irrelevant in this research because it is related to a very small percentage of the orders in this study. The delivery conditions are also eliminated from further research because the majority of the studied orders have the same conditions.

Using regression analysis, useless (statistically insignificant) and non-influential columns of information were selected. Regression analysis is used to investigate the relationship between two or more variables. The primary objective of regression analysis is to examine how a dependent variable is affected by one or more independent variables. It helps to estimate the strength and direction of the relationship between the variables and make predictions about the dependent variable based on the values of the independent variables [24]. The selection criteria were based on such small data (29 production orders), and the confidence level was set to *p* < 0.05. Based on this, it was found that the quantity of the order, type of payment, customer rating, and order span had the greatest influence on profit. Firstly, the most profitable orders were planned, then, the other orders were launched if they had the materials, machines, and workers available.

#### 3.1.1. Materials

In the manufacturing process, three primary components were utilized to fulfill the production orders, namely, materials, machinery, and employees. The materials required for each product are outlined in Table 2, and the developed methodology assesses whether the necessary quantities are available. In instances where orders were stalled due to a shortage of materials, other orders were given priority to prevent further delays. As in this research, each product had several different raw materials because most of the products were bent, welded, metal frames, as shown in Figure 4. For example, such a product was produced from the R5 cold rolled steel bar and laser-cut metal plates.

#### 3.1.2. Machinery

Similar to the process for the materials, it is imperative to conduct a thorough examination of the machinery to ensure its operational capacity. This research involved analyzing production data to determine the functionality of each machine assigned to specific operations. Table 5 provides an overview of the number of available machines for each operation.

Since information is available on how long each operation takes, it is easy to calculate when the machine will be free and schedule the start of the next operation. An ongoing operation can be stopped only if a more profitable order appears unless the originally started order needs to be completed due to deadlines.

#### 3.1.3. Employees

This sub-subsection includes the previously published article information about employee replacement blocks [2]. It describes the steps in employee reconfiguration owing to an absence. The most important part of the research was the matrix of skills, which is used in this method, as well as the additional improvements.

In this research, the key point was to optimize production from the perspective of the biggest possible profit. Thus, additional information, such as the hourly wage of employees or the hourly price of machinery must be involved. Employee hourly wage is presented in Table 6, and it differs based on each employee’s skills and knowledge. Table 7 presents the machinery hourly price—depending on the energy consumption of machinery, its deterioration, created value, etc.

## 4. Results

In this research, the length of a working day’s shift was divided by half an hour (considering the shortest possible duration of the order). For this purpose, a matrix for orders is created. Data in it are divided by half an hour. Each 30 min can represent a different operation and the order of operations can be changed every 30 min. As an example, one shift working plan is presented in Figure 5. Colors mark different operations from Table 5. This shift works on several orders but from the chart, it can be seen that many orders are not in a process.

Additionally, the matrix of the performed tasks for each working employee is created. It represents the operation being performed by each employee and which employee has no task—white spaces in Figure 6.

In this paper, real-life cases are researched, and production plans consist of 29 previously mentioned production orders. In total, 18 employees were working in 1 shift, and 13 operations can be performed. The first check was when optimization was not performed. If production orders follow input order date, after 16 working hours, the working plan will be as presented in Figure 7a, and the operations of the employees will be as in Figure 7b.

As seen in Figure 7a, only a few orders are performed after 16 h; thus, the machinery and employees are not sufficiently loaded, as shown in Figure 7b. Based on that, optimization should be performed to improve these results. The present study involves the computation of the duration for each order, in accordance with the associated operations. Subsequently, the average duration of the orders was determined, and the orders with a duration exceeding the average were fragmented into smaller units. This fragmentation strategy ensures that the order duration does not surpass the computed average, and that the allocation of operations to employees is proportionate. The net outcome of this strategy was a rise in the number of orders to 44, as large orders were disintegrated into smaller units. The findings presented in Figure 8 demonstrate that although there was an improvement in the number of running orders after 16 h, the figures remained suboptimal (Figure 8a). The optimization measures resulted in increased task assignments for the employees (Figure 8b).

A further round of optimization was performed, whereby the production orders were prioritized based on several critical factors. Notably, the order quantity, payment type, customer rating, and order duration were identified as having the most significant impact on overall profitability. Therefore, these factors were used to rank and schedule the production orders for enhanced profitability. Based on the latest findings, after this optimization, it was evident that the number of active production orders remained relatively constant (Figure 9a). However, there was a notable increase in the diversity of tasks performed by employees, resulting in a higher workload for the machinery (Figure 9b). Despite this, it is noteworthy that employees are not operating at their maximum capacity, suggesting that the current workforce may not be optimally aligned with the available machinery resources.

Following an analysis that identified some employees who were deemed surplus to requirements, a third round of optimization was undertaken. Specifically, three employees were selected for removal based on their limited ability to perform high-quality tasks. These were employees 6, 8, and 9. Subsequently, a revised work plan was formulated, detailing working intervals across all shifts (0–8 h, 8–16 h, and after 16 h). The first shift is presented in Figure 10. The second shift is presented in Figure 11, and the last shift, which was analyzed in pictures before in this section, is presented in Figure 12.

Following the optimization steps outlined, the total production time for all orders has been reduced. Specifically, the total time for the production of these 29 orders decreased from 47 to 42 h. A visual comparison of the production situation before optimization, where the total time was 47 h, and the situation after the third round of optimization, which resulted in a 10% reduction in time, can be observed in Figure 13 and Figure 14, respectively.

After undergoing a three-stage optimization process utilizing the developed methodology, it has become evident that the current production situation requires modifications. This methodology not only provides immediate solutions for adapting production but also offers guidance on reorganizing production for sustained improvement.

The experimental results of DSM DPP testing reveal that introducing additional machinery is imperative to improve the production situation. Cutting, being a fundamental operation required for most orders, serves as a bottleneck in the production process, and hence, any enhancements made to this process would be particularly advantageous. This operation acts as a prerequisite for subsequent operations, and any delays in this process result in a direct impact on order flow. Additionally, finishing is another operation that is necessary for the majority of orders, and it takes, on average, two to three times longer than other operations. Therefore, streamlining this operation represents another critical step toward optimizing the order flow. Additionally, the concept of exploring the potential subcontractors as a possible solution could provide an expedient resolution to the current situation.

Further, investing in employee training, particularly those with limited skill sets, is another recommendation that should be considered. Providing additional training to employees can diversify their skill sets, enabling them to cover a broader range of workloads. Failure to provide this training may result in operational inefficiencies, as evidenced by the removal of three employees, which resulted in little disruption to the overall workflow.

These brief notes serve as an initial exploration into the effectiveness of the proposed method. It is important to note that conducting longer experiments would yield more precise and accurate insights into the method’s efficacy.

## 5. Discussion

Even though this method was originally created based on the needs of an SME-type manufacturing company, it is possible to adapt it to a different field—for companies with no production but service providers. As for the future, a new study with the automotive body repair company was started and implementation of the method went smoothly. Some of the calculations have already shown promising results and this presents the possibility of this method to be universal. However, after research in this field, it can be stated that this method might not be so relevant for the companies out of the SME field—large-scale companies mostly have working ERP systems, which cover the presented issues. Such companies can even have automatic production lines, which eliminates most of the problems regarding employees and their skills. Furthermore, this method might not be so necessary for SME companies, which have batch production, and their manufacturing processes are not so dissected.

## 6. Conclusions

The industrial sector faces continuous evolution in the production processes, yet production issues and delays are still common. This article concentrated particularly on small and medium-sized enterprises (SMEs), which often struggle to implement the latest Industry 4.0 technologies. While these technologies can provide advanced solutions, the reliance on human expertise in such companies is still very high and can lead to delays and inefficiencies in the production process. The article highlights the importance of implementing Industry 4.0 technologies and especially those that can ensure real-time data and could be adapted without large investments—the Industrial Internet of Things, blockchains, sensors. To address this issue, a decision support method for dynamic production planning was developed, which is based on real-time data and involves algorithms and programming in Matlab to optimize the process. A case study in a metal processing company was completed to test this method. After a three-stage optimization, it was confirmed that the company can save up to 10% in production time. This would lead to a reduction from 47 to 42 working hours to perform the scheduled tasks. In total, 29 production orders were conducted during the test, with a total of 18 employees, and 13 operations. The method is applicable to different production settings, demonstrating its potential to improve production processes and increase profitability, particularly in SMEs. For larger-scale companies or SMEs with batch orders, this method might not be so relevant, although adaptability is possible. Using this method, a company can create long-term plans based on the results. A few insights on the long-term improvements were provided after this specific case study. Combining real-time responses and replanning with future improvements will lead to successful and efficient businesses.

## Figures and Tables

**Figure 1 sensors-23-04498-f001:**
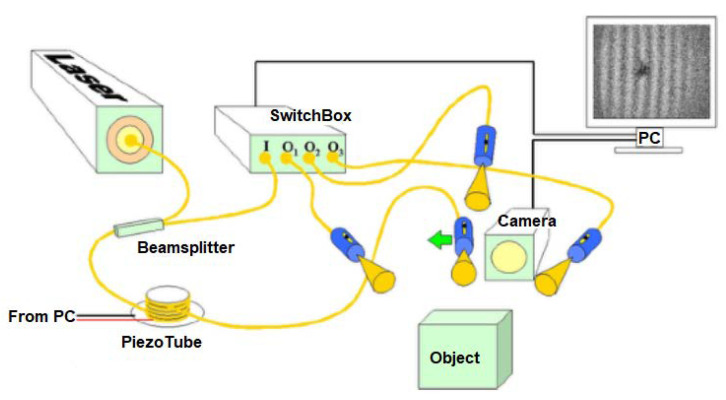
Optical device for measuring complex structures [8].

**Figure 2 sensors-23-04498-f002:**
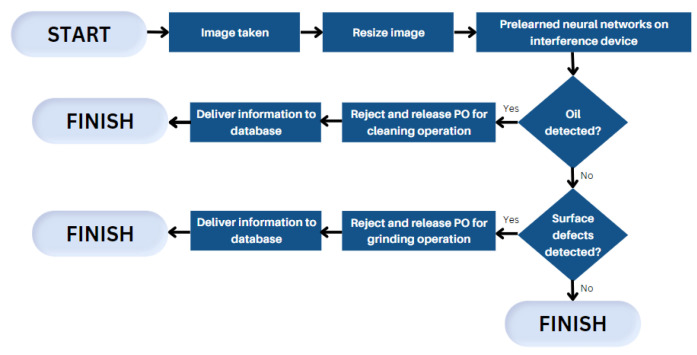
Flow chart of the operational process.

**Figure 3 sensors-23-04498-f003:**
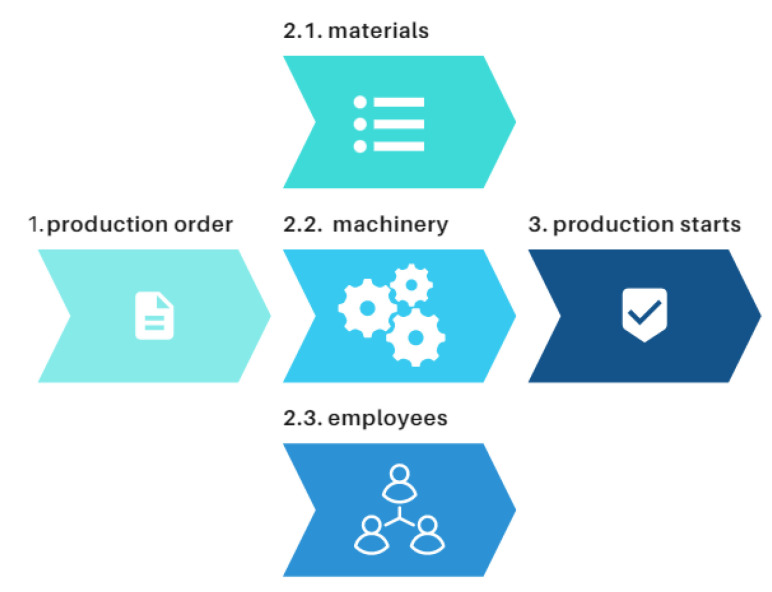
Sequence in production.

**Figure 4 sensors-23-04498-f004:**
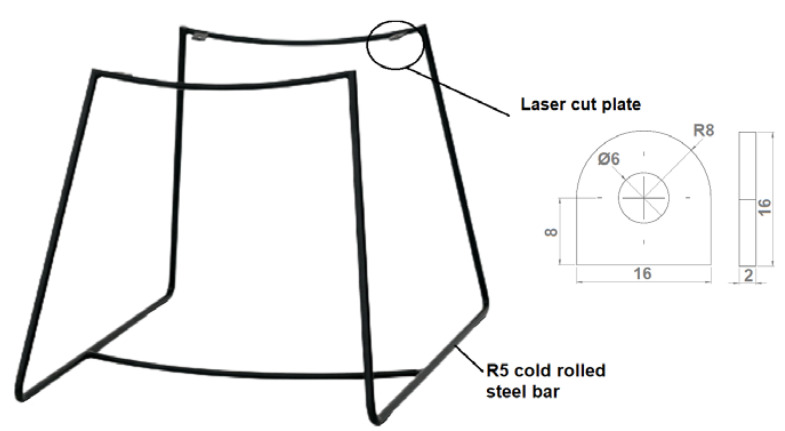
Product—metal frame.

**Figure 5 sensors-23-04498-f005:**
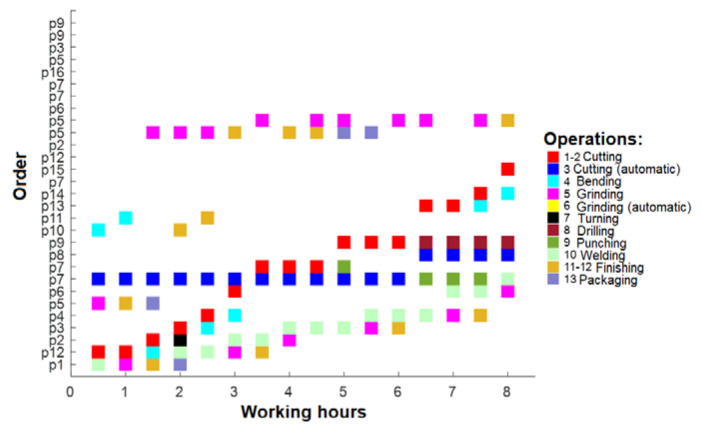
One shift working plan.

**Figure 6 sensors-23-04498-f006:**
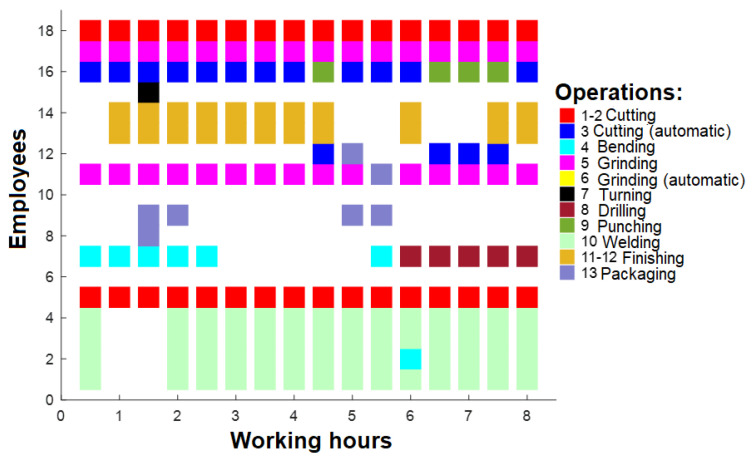
Operations are divided by employees.

**Figure 7 sensors-23-04498-f007:**
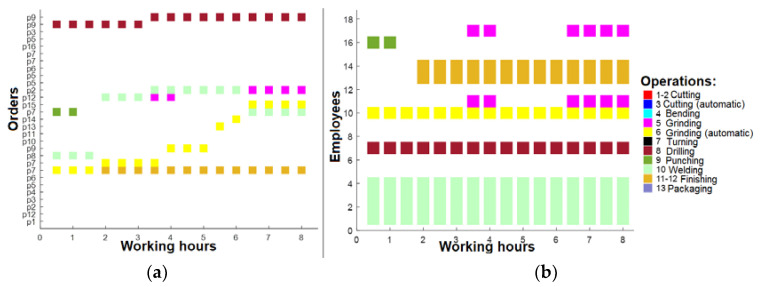
(**a**) Working plan after 16 h; (**b**) operations for employees after 16 h.

**Figure 8 sensors-23-04498-f008:**
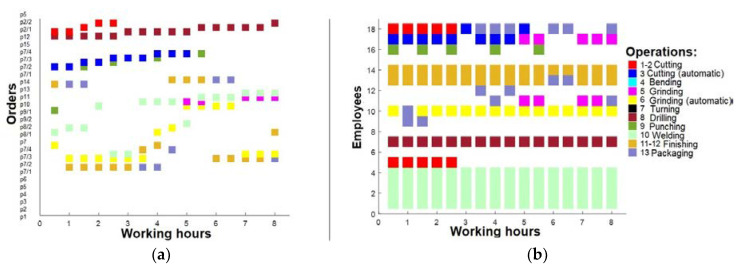
(**a**) Working plan after 16 h (after 1st optimization); (**b**) operations for employees after 16 h (after 1st optimization).

**Figure 9 sensors-23-04498-f009:**
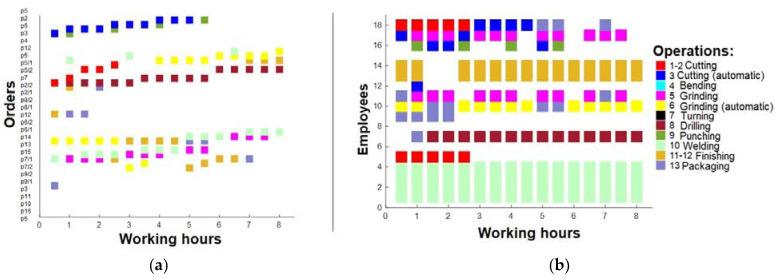
(**a**) Working plan after 16 h (after 2nd optimization); (**b**) operations for employees after 16 h (after 2nd optimization).

**Figure 10 sensors-23-04498-f010:**
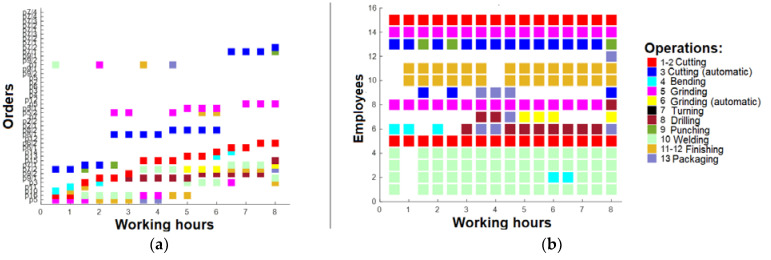
(**a**) Working plan for 1st shift (after 3rd optimization); (**b**) operations for 1st shift employees (after 3rd optimization).

**Figure 11 sensors-23-04498-f011:**
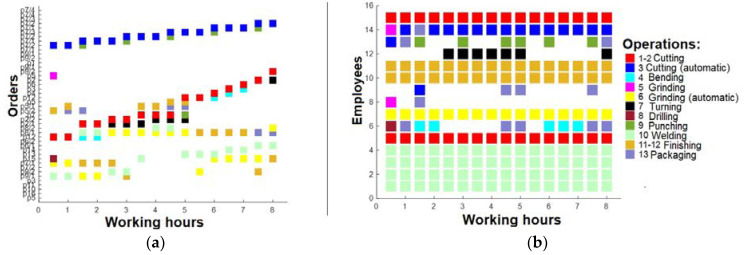
(**a**) Working plan for 2nd shift (after 3rd optimization); (**b**) operations for 2nd shift employees (after 3rd optimization).

**Figure 12 sensors-23-04498-f012:**
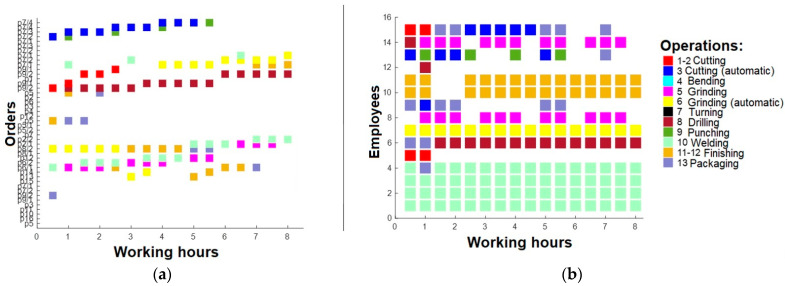
(**a**) Working plan for 3rd shift (after 3rd optimization); (**b**) operations for 3rd shift employees (after 3rd optimization).

**Figure 13 sensors-23-04498-f013:**
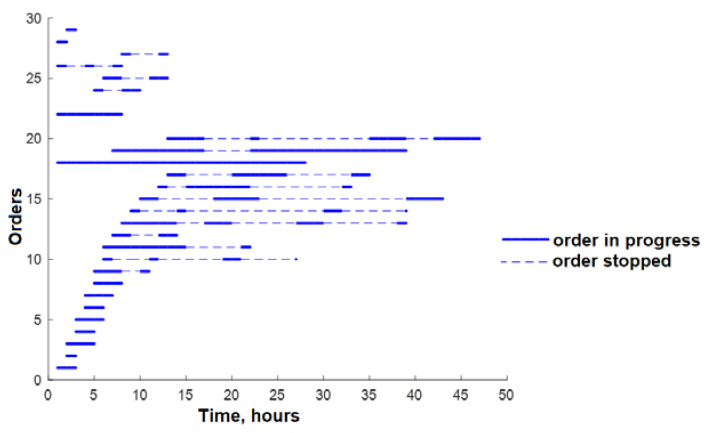
Production time-lapse for 29 orders with no optimization.

**Figure 14 sensors-23-04498-f014:**
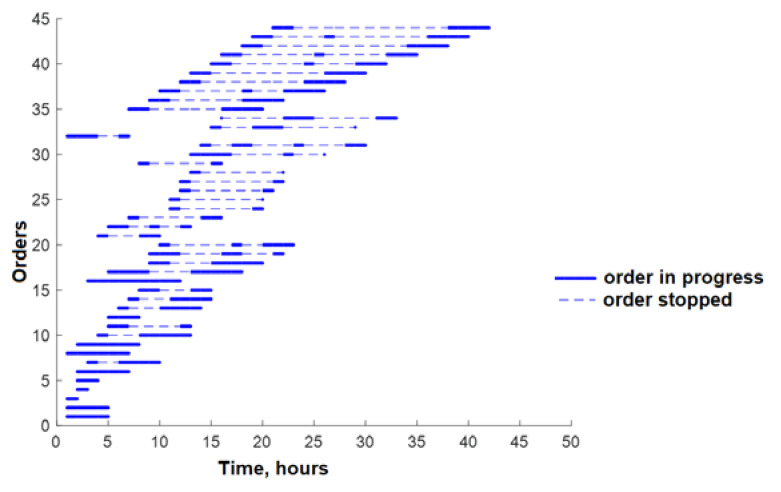
Production time-lapse for divided 29 orders with three-stage optimization.

**Table 1 sensors-23-04498-t001:** Initial data for production planning.

Customer	Order No.	Product No.	Quantity	Order Span, Days	Order Date	Value, Euro	Value of Production, Euro	Delivery Type	Payment Type	Customer Rating	Delay Ratio	New Product	Shipping	Complexity of Product	Rejection Ratio	Subcontractors	Time in Subcontractors, Days	Materials
A	1	P1	1	28	12/12/2022	186	86	0	30	0.8	50	1	1	0.5	0	0	0	1, 2
A	2	P12	1	21	12/7/2022	231	197	0	30	0.8	50	0	1	0.7	10	0	0	1, 2
A	2	P2	1	21	12/7/2022	145	74	0	30	0.8	50	0	1	0.7	10	0	0	2, 3, 4
A	2	P3	1	21	12/7/2022	296	141	0	30	0.8	50	0	1	0.5	0	0	0	2, 3
A	2	P4	1	21	12/7/2022	304	156	0	30	0.8	50	0	1	0.3	0	0	0	2, 3
A	2	P5	1	21	12/7/2022	171	65	0	30	0.8	50	0	1	0.2	0	1	4	1
A	2	P6	1	21	12/7/2022	245	100	0	30	0.8	50	0	1	0.4	0	0	0	2
B	3	P7	720	35	11/29/2022	4320	2265	0	30	0.7	80	0	1	0.3	10	0	0	2, 3
C	4	P7	100	21	12/14/2022	700	365	50	30	0.7	65	0	1	0.3	10	0	0	2, 3
C	4	P8	200	21	12/14/2022	1280	780	0	30	0.7	65	0	1	0.2	10	0	0	2
D	5	P9	300	10	11/25/2022	438	151	0	30	0.9	25	0	1	0.4	10	0	0	2
E	6	P10	10	10	11/2/2022	21	11	0	−1	0.6	50	1	1	0.2	0	0	0	4
…																		

**Table 2 sensors-23-04498-t002:** Data on operations and time.

Customer	Operations	Time of Operation for 1 Piece (min)										
		1, 2	3	4	5	6	7	8	9	10	11, 12	13
A	10, 5, 11, 13				30					120	60	12
A	1, 2, 4, 10, 5, 11, 13	45		30	25					130	45	9
A	1, 2, 7, 10, 5, 11, 12, 13	10			20		15			35	40	9
A	1, 2, 4, 10, 5, 11, 12, 13	30		20	20					75	25	15
A	1, 2, 4, 10, 5, 11, 13	30		20	20					75	35	15
A	5, 11, 13				30						30	15
A	1, 2, 10, 5, 11, 13	15			30					60	40	18
B	3, 9, 10, 6, 11, 12, 13		360			480			72	480	920	216
C	1, 2, 9, 10, 6, 11, 12, 13	100				120			20	200	60	60
C	3, 10, 6, 11, 12, 13		200			360				300	420	240
D	1, 2, 8, 10, 6, 11, 12, 13	150				75		300		140	50	90
E	4, 11, 13			30							5	6

**Table 3 sensors-23-04498-t003:** Description of columns.

Column	Short Description
Customer	This column presents customers’ names (in this research, based on privacy rules, names are marked as A, B, C, etc.)
Order no.	This column presents order number. In one order several different products could be ordered.
Product no.	This column presents unique product number.
Quantity	The number of pieces per order.
Order span, days	Time in days from order confirmation to delivery date.
Order date	Date when the order was confirmed.
Value, euro	Total value received from customer of the specific product and quantity of the order in euros.
Value of production, euro	The amount of money left after raw materials value is taken away in euros.
Delivery type	Customers can agree to get a partial delivery—divide the order into several pieces. In this column, a percentage of minimum required order quantity is given.
Payment type	Customers might pay in advance (value “−1”) or have postponed payment (30, 60 days).
Operations	Operations must be done in the correct order and this column represents which operation is needed and when each of it could be done. Meanings of operations are described in Table 4.
Time of operation for 1 piece, minutes	Minutes for each operation for one piece of product.
Customer rating	Each customer is ranked based on several individual aspects—the percentage of its order compared with total orders in company, payment in time ratio, specific agreements, etc.
Delay ratio	The percentage of delivered late orders from all previous orders.
New product	If the product is new, the value is “1”. If the product was produced previously, the value is “0”.
Shipping	Order might be delivered at place (DAP), or customer should organize transport or pay for it when it is Ex Works (EXW) conditions. DAP means that all specified order span with shipping included (which can take several days).
Complexity of product	Scale to 1—the bigger values, the more complex the product and the time spent on production is longer.
Rejection ratio	Percentage of how many products were rejected in previous production orders.
Subcontractors	If the product need operations made by other companies, the value is “1”. If the product is made only in this company, the value is “0”.
Time in subcontractors, days	If subcontractor is needed, the time required for it is provided.
Materials	Each product has specifications of what raw materials are needed and list of what is required. Here to simplify data, materials are coded.

**Table 4 sensors-23-04498-t004:** Operations for the specific research.

Operation No.	Operation Name
1	Cutting
2	Cutting
3	Cutting
4	Bending
5	Grinding
6	Grinding
7	Turning
8	Drilling
9	Punching
10	Welding
11	Finishing
12	Finishing
13	Packaging

**Table 5 sensors-23-04498-t005:** Machinery of each operation.

Operation No.	Used Machine for the Operation	Quantity
1	Band saw machine	1
2	Disc saw machine	1
3	Automatic disc saw machine	1
4	Bending machine	1
5	Manual sander	2
6	Belt sander	1
7	CNC turning machine	1
8	Vertical milling machine	1
9	Hydraulic press	1
10	Welding machine	4
11	Powder coating booth (3 × 1, 54 × 1, 8 m)	1
12	Powder coating booth (2 × 1, 1 × 1, 8 m)	1
13	Hand saw	1

**Table 6 sensors-23-04498-t006:** Hourly wage of employees.

Employee No.	Hourly Wage (Euros)
1	9.25
2	9.5
3	9.5
4	9
5	4.7
6	4.5
7	4.5
8	4.75
9	4.5
10	4.75
11	4.75
12	4.75
13	7
14	7
15	6
16	4.75
17	4.75
18	4.5

**Table 7 sensors-23-04498-t007:** Machinery hourly price.

Operation No.	Hourly Price (Euros)
1	5
2	3
3	3
4	3
5	1
6	5
7	10
8	2
9	1
10	7
11	20
12	15
13	0.5

## Data Availability

Not applicable.

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
