# Peer review of "Optimization Experiment of Production Processes Using a Dynamic Decision Support Method: A Solution to Complex Problems in Industrial Manufacturing for Small and Medium-Sized Enterprises"

_sensors, 2023, doi:10.3390/s23094498_

Round 1

Reviewer 1 Report

The author uses dynamic production planning (DSM DPP) to solve Complex Problems in Industrial Manufacturing for Small and Medium-sized Enterprises. In general, the article is interesting, and the author has conducted a large number of experiments to verify it. I mainly have the following problems:

1. What is the motivation for the DSM DPP approach? Its core idea and algorithm flow should be given in detail.

2. What are the methods to solve the above problems? What are the advantages and disadvantages of these methods? What are the core advantages of DSM DPP algorithm?

3, the author should come up with more algorithm comparison to illustrate the advantages of the algorithm proposed in this paper.

4. Whether this method has universality or limitations in other fields. For example, whether it can be applied in large enterprises.

  • Language writing in general is OK.

Author Response

Dear Reviewer 2,

We strongly appreciate provided review report. All of the comments were considered and changes were made. 

Below is point to point explanation of changes that were made:

  1. The abstract and introduction are now filled with more precise aim of this topic - to show the effectiveness of created method which evaluates not only technological but also human factors.
  2. Based on this point, additional DSS were examined in Introduction section to prove that it is not much research for dynamic production planning where human factor is involved - skills of operations, speed of working or salary size. The main advantage and novelty from our method is clearly not so much research in other such type methods.
  3. We assumed that this Point is connected to Point no.2 so we added valuable and recently published research based on this topic to prove difference from other decision support systems - our algorithm evaluates include employees. 
  4. In Discussion section we added information about limitations of this method and explained to which type of companies this could be most relevant. 

In overall, we appreciate all remarks and we tried to briefly response to all of them. We see that after implementing these reviews, article gained additional strength. 

Thank you in advance.

Reviewer 2 Report

Thank you for the opportunity to read and review this paper. The complex problems of the Industry 4.0 and advanced technological solutions is relevant in practice. The article "Optimization experiment of Production Processes using Dynamic Decision Support Method: A Solution to Complex Problems in Industrial Manufacturing for Small and Medium-sized Enterprises" was concentrated particularly on small and medium-sized enterprises (SMEs) that often struggle to implement the latest Industry 4.0 technologies. The article develops a decision support method for dynamic production planning, which is based on real-time data and includes algorithms and programming in Matlab to optimize the process.

The work is interesting, to highlight different aspects:
- The problems and  data collected is interesting.
- The problems are clearly stated.
- The theoretical framework are creative.
- The methodology is clearly explained.  
- The study conclusions supported are by the analysis.
My following comments are intended to be constructive and hope they are helpful to the authors:
Abstract - authors should this section changed. The abstract should have to provide more structured aim and scope, to state the principal objectives and scope of the investigation and describe the paper's originality and value. The innovation of the paper is  missing.
Introduction - please briefly describe in the last paragraph of the INTRODUCTION section, the content of each section of the paper (Section 1-4), as well as include brief information on methods (one sentence).
Discussion: authors should this section changed. This section must compare obtained results with other authors.

Conclusions: authors should this section changed. This section should emphasis the objective and main results presented in this study. The conclusions must concisely summarise the main points of the paper.
This part usually includes four compulsory elements:
(1) general summary of the article, its results and findings,
(2) implications and recommendations for practice,
(3) research limitations, and
(4) suggestions for future research.

Author Response

Dear Reviewer 3,

we are truly thankful for your review. We kindly considered all of your remarks and implemented changes. Below is point to point update to your review:

Abstract - we improved by adding more precise aim of this research. As well we added examples to prove novelty of our method.
Introduction - sections described in last paragraph.
Discussion - we separated this section from conclusions and added data about our methods limitations. Examples of other authors were added in introduction to add more points in the beginning. 
Conclusions - we separated part of information from conclusions to discussions. This made conclusions more concentrated on the results of the research. 

We truly appreciated all of the comments and agreed to them. 

Thank you in advance. 

Round 2

Reviewer 1 Report

I suggest accept the paper.